# Enhanced Reactive Brilliant Blue Removal Using Chitosan–Biochar Hydrogel Beads

**DOI:** 10.3390/molecules28166137

**Published:** 2023-08-19

**Authors:** Yangyang Zhao, Yang Song, Rui Li, Fengfan Lu, Yibin Yang, Qiongjian Huang, Dongli Deng, Mingzhu Wu, Ying Li

**Affiliations:** 1Chemical Pollution Control Chongqing Applied Technology Extension Center of Higher Vocational Colleges, Chongqing Industry Polytechnic College, Chongqing 401120, China; songyang@cqipc.edu.cn (Y.S.); luff@cqipc.edu.cn (F.L.); yangyb@cqipc.edu.cn (Y.Y.); huangqj@cqipc.edu.cn (Q.H.); dengdl@cqipc.edu.cn (D.D.); wumz@cqipc.edu.cn (M.W.); 2School of Biological Science, Jining Medical University, No. 669 Xueyuan Road, Donggang District, Rizhao 276826, China; ruili061289@163.com

**Keywords:** biochar, chitosan, reactive brilliant bule, adsorption, mechanism

## Abstract

To address the challenges associated with the weak affinity and difficult separation of biochar, we developed chitosan–biochar hydrogel beads (CBHBs) as an efficient solution for removing reactive brilliant blue (RBB KN-R) from wastewater. The adsorption behavior and mechanism of RBB KN-R onto CBHBs were extensively studied. Notably, the adsorption capacity of RBB KN-R showed pH-dependence, and the highest adsorption capacity was observed at pH 2. The adsorption process was well fitted with the pseudo-second-order kinetic model and the intraparticle diffusion model. Film diffusion and intraparticle diffusion were both responsible for the adsorption of RBB KN-R onto CBHBs. At 298.15 K, the maximum adsorption capacity *q*_m_ was determined to be 140.74 mg/g, with higher temperatures favoring the adsorption process. A complex mechanism involving π–π interactions, electrostatic attraction, hydrophobic interaction, and hydrogen bonding was found to contribute to the overall adsorption process. The experimental data discovered the coexisting substances and elevated ionic strength hindered the adsorption capacity. Significantly, after three cycles of adsorption–desorption, the CBHBs maintained an adsorption capacity above 95% for RBB KN-R. These promising results imply that CBHBs are a durable and cost-effective adsorbent for efficient removal of dyes from wastewater.

## 1. Introduction

The printing and dyeing industry is a major contributor to the water pollution as it discharges large amounts of dyes into wastewater [1,2]. Approximately 70 million tons of dyes are produced and used annually worldwide [3]. The presence of excessive dyes in wastewater poses significant risks to plants, aquatic organisms, and human health. Adverse effects on the human body include impacts on the brain, kidneys, heart, respiratory system, and immune system [4]. Additionally, many dyes have been identified as mutagenic and carcinogenic [5]. Reactive brilliant blue (RBB KN-R) is an anthraquinone dye widely employed in the printing and dyeing industry [6]. High concentration of RBB KN-R can impede photosynthesis, leading to plant death and ecological contamination.

Numerous methods, including oxidation, biodegradation, electrochemical degradation, membrane filtration, are available to treat dye wastewater [7,8]. However, adsorption has emerged as an economical and effective technology for dye removal from water due to its high efficiency, ease of operation, and reusability [9,10]. Among the various adsorbents, biochar (BC) has garnered increasing attention because of its environmentally friendly nature and cost-effectiveness.

BC is a carbon-rich solid material, which is obtained through the thermal treatment of biomass under oxygen-free or low-oxygen conditions. The biomass used for its production primarily consists of agricultural residues, animal waste, and seed husks [11,12]. Multiple methodologies are implemented in the production of biochar, including slow pyrolysis, fast pyrolysis, gasification, microwave-assisted pyrolysis, hydrothermal carbonization, flash carbonization, torrefaction, and gasification. Among these methods, slow pyrolysis emerges as the predominant approach, characterized by heating rates between 1 to 30 °C/min and a maximum temperature below 700 °C [13,14]. With its high porosity, large surface area, and abundant functional groups, BC has gained increasing attention in wastewater treatment applications. However, the utilization of BC is constrained as for its weak affinity, low density, tendency to aggregate, and difficulty in separation from water [12]. In order to overcome those referred shortcomings, researchers have explored the encapsulation of BC within polymer materials to enhance its adsorption performance and recyclability [15,16].

Chitosan, an N-deacetylated derivative of chitin, is a linear polymer of acetamidos-D-glucose and ranks as the second most abundant biological resource [17]. This material offers several advantages, including cost-effectiveness, nontoxicity, natural antibacterial properties, and biodegradability [18]. Currently, chitosan has attained extensive application in the wastewater treatment. However, the low mechanical strength, weak stability, and limited surface area of raw chitosan particles restrict their application [17]. Therefore, chitosan commonly engages in the formation of various composites with diverse materials to facilitate the adsorption performance and recycling process of the adsorbents, while BC exhibits great potential for the sorption of various pollutants due to its large surface areas and numerous functional groups [19]. Therefore, the synergistic combination of chitosan and BC can improve the mechanical strength and stability of chitosan and enhance the recycling capability of BC.

In this study, composite chitosan–BC hydrogel beads (CBHBs) were fabricated by encapsulating BC within chitosan to enhance stability, adsorption performance, and recycling ability for the removal of RBB KN-R. Rice husk was subjected to pyrolysis to produce BC. The properties of CBHBs were characterized, and the adsorption performance and removal efficiencies of CBHBs for RBB KN-R under various conditions were investigated. Furthermore, adsorption mechanisms were proposed using diverse characterization methods. The findings of this research are anticipated to provide valuable insights for synthesizing BC composites and effectively treating dye-contaminated wastewater.

## 2. Results and Discussion

### 2.1. Characterization Results

Figure 1 displays the SEM images of BC and CBHBs before and after adsorption. As observed in Figure 1d, the CBHBs were successfully fabricated with a grain size of 924.2 μm. Moreover, as depicted in Figure 1a–d, the surface of BC appeared smooth, whereas the surface of CBHBs exhibited roughness and the presence of numerous pore channels. The CBHBs exhibited a measured BET surface area of 26.5 m^2^/g, encompassing a total pore volume of 0.067 cm^3^/g, along with an average pore width of 8.4811 nm (refer to Appendix A). These findings provide unequivocal evidence for the existence of internal mesopores within CBHBs. Following the adsorption process, depicted in Figure 1e,f, the pores and surface of CBHBs were filled with RBB KN-R. These results demonstrated that CBHBs were successfully used in the adsorption of RBB KN-R.

The analysis of functional groups on the surface of the BC and CBHBs was performed using FTIR spectra, and the results are presented in Figure 2. As observed from the FTIR spectra, a noticeable change was found between the BC (a) and CBHBs before adsorption (b). For BC, the FTIR spectrum was assigned to the –OH stretching and amine N–H stretching (3430 cm^−1^) [20], C=O stretching vibration band (1711 cm^−1^) [21], benzene ring skeleton vibration and C=C stretching vibration (1575 cm^−1^) [22], C–H bending vibration (1416 cm^−1^) [23], and C–O stretching vibration (1000–1300 cm^−1^). For CBHBs before adsorption (b), the wider and stronger –OH stretching and amine N–H stretching at around 3430 cm^−1^, the new C–O–C at around 1078 cm^−1^, and the amide I C=O stretching vibration at around 1642 cm^−1^ all confirmed that the chitosan had been crosslinked with biochar [24], indicating the successful synthesis of CBHBs. Overall, both BC and CBHBs exhibited abundant functional groups on their surfaces, rendering them excellent adsorbents.

### 2.2. Effect of Solution pH

The influence of solution pH on the adsorption process is of utmost significance due to its impact on both the degree of RBB KN-R speciation and the dissociation of functional groups in CBHBs [25]. To investigate the adsorption capacity of RBB KN-R on CBHBs across a wide pH range, a series of experiments were conducted, and the results are displayed in Figure 3. The data presented in Figure 3a revealed that the highest *q*_e_ value occurred at pH 2, followed by a decreasing trend as the pH increased. Interestingly, a slight increase in *q*_e_ was observed around pH 7. Additionally, Figure 3b demonstrates that the pH_PZC_ of CBHBs was approximately 7.2.

In an aqueous solution, RBB KN-R demonstrated a pKa value below 1, resulting in the predominance of negatively charged dye anions (Dye-SO_3_^−^) when the pH surpassed 1. When the pH was lower than the pHpzc, CBHBs carried a positive net charge, leading to enhanced electrostatic interaction between Dye-SO_3_^−^ and CBHBs. Consequently, the adsorption capacity showed an elevated value. With decreasing pH, the net charge of CBHBs increased, resulting in higher adsorption capacity. Conversely, when the pH exceeded the pHpzc, CBHBs acquired a negative charge, leading to electrostatic repulsion between Dye-SO_3_^−^ and CBHBs. Moreover, OH^-^ ions in the solution competed with Dye-SO_3_^−^ for binding sites on the surfaces of CBHBs, thereby reducing the adsorption capacity [26]. These findings strongly suggested that electrostatic attraction significantly contributed to the adsorption mechanism.

Near the pH_PZC_ of CBHBs (pH 7.2), where the net charge of CBHBs and the electrostatic interaction between Dye-SO_3_^−^ and CBHBs were negligible, other factors, such as π–π interactions, hydrophobic interactions, and hydrogen bonds facilitated by the benzene ring and –NH_2_ on RBB KN-R, collectively contributed to the enhancement of RBB KN-R adsorption onto CBHBs. This enhanced adsorption capability ultimately led to an increased adsorption capacity [26]. Based on these outcomes, pH 2 was selected for subsequent studies on RBB KN-R removal.

### 2.3. Comparison of BC and CBHBs

To evaluate the efficiency of the synthesized CBHBs, the removal performance of BC and CBHBs for RBB KN-R was investigated. The findings are presented in Figure 4a. As depicted in Figure 4a, CBHBs exhibited significantly higher removal efficiency for RBB KN-R compared to BC. This observation highlighted the effectiveness of the synthesized CBHBs and the synergistic combination of chitosan and BC in their composition.

### 2.4. Effect of CBHBs Dosage

The dosage of CBHBs plays a significant role in determining the degree of RBB KN-R removal. To investigate the impact of CBHBs dosage, experiments were conducted to measure the RBB KN-R removal efficiency at varying CBHBs dosages ranging from 0.2 g to 1.8 g. The corresponding results are illustrated in Figure 4b. It can be observed from the figure that the RBB KN-R removal efficiency increased proportionally with the increment of CBHBs dosage. This finding aligned with earlier studies conducted on methylene blue removal [27], as well as the tetracycline and doxycycline removal [28]. The enhanced removal efficiency can be attributed to the enlarged adsorption surface area and increased availability of adsorption sites resulting from higher CBHBs dosages.

### 2.5. Adsorption Kinetics

The influence of contact time on the adsorption of RBB KN-R onto CBHBs at different initial concentrations (50 mg/L and 100 mg/L) was investigated, and the results are depicted in Figure 5. As demonstrated in Figure 5, there was a rapid initial increase in adsorption capacity within the first 2 h, followed by a stable period until reaching adsorption equilibriums. The rapid increase in adsorption during the initial phase was attributed to the high concentration of RBB KN-R and the availability of numerous adsorption sites on the surface of CBHBs. Subsequently, the stability in adsorption capacity was a result of the lower concentration of RBB KN-R and the occupation of adsorption sites [29,30].

To analyze the adsorption kinetics, the experimental data were fitted using the pseudo-first-order kinetic model (Figure 5a), pseudo-second-order kinetic model (Figure 5b), and intraparticle diffusion model (Figure 5c). The corresponding fitting parameters are presented in Table 1. Based on the correlation coefficient (R^2^) values in Table 1, the pseudo-second-order kinetic model provided a better fitting compared to pseudo-first-order kinetic model, whereas we believe that both physical and chemical reactions were involved in the adsorption process. Likewise, Zhang also found that Cr (VI) adsorption onto chitosan-modified magnetic biochar was physically and chemically affected by the porous structure and functional groups within the adsorbent with a better pseudo-second-order kinetic model fitting [21]. Additionally, the application of the intraparticle diffusion model (Figure 5c) revealed that the adsorption process consisted of three steps: film diffusion, intraparticle diffusion, and adsorption equilibrium [31,32]. However, the regression lines did not pass through the origin, indicating that both film diffusion and intraparticle diffusion contributed to the adsorption of RBB KN-R [32].

### 2.6. Adsorption Isotherms and Thermodynamics

The adsorption isotherms of RBB KN-R onto CBHBs were investigated as a function of initial RBB KN-R concentration (0–300 mg/L) at different temperatures (298.15 K, 308.15 K, and 318.15 K), and the results are presented in Figure 6. To assess the adsorption behavior, the Langmuir isotherm (Figure 6a) and Freundlich isotherm (Figure 6b) models were employed, and the corresponding parameters were determined and are listed in Table 2. As depicted in Figure 6, the adsorption capacity increased with increasing initial RBB KN-R concentration until reaching equilibrium.

The Langmuir isotherm suggests that adsorption takes place at specific, homogeneous sites on the adsorbent surface, while the Freundlich isotherm suggests that adsorption occurs at heterogeneous sites with uneven distribution of affinity and heat of adsorption on the surface [33,34]. Clearly, the adsorption data exhibited better fitting with the Freundlich isotherm, as they displayed a higher R^2^ value, indicating a nonuniform surface of CBHBs with multiple layers of adsorption. According to the Langmuir isotherm, the maximum monolayer adsorption capacity (*q*_m_) for RBB KN-R was determined as 140.74 mg/g at 298.15 K, 140.98 mg/g at 308.15 K, and 143.38 mg/g at 318.15 K. The obtained *R*_L_ values ranged from 0 to 1, signifying favorable adsorption. According to the Freundlich isotherm, the calculated values of 1/n, ranging between 0 and 1, further supported favorable adsorption. It is anticipated that energy was significantly reduced upon coverage of binding sites.

Table 3 depicts the adsorption capacity of alternative adsorbents for RBB KN-R. As evident in Table 3, a diverse range of adsorbents was employed for the elimination of RBB KN-R, with biochar alone typically exhibiting lower adsorption capacity than composites [26,35]. Notably, CBHBs demonstrated a favorable adsorption capacity compared to other adsorbents. Taking into account the environmentally friendly characteristics and facile synthesis of CBHBs, it can be inferred that CBHBs offered an effective solution for the removal of RBB KN-R as an adsorbent.

The effect of temperature on adsorption was also examined, and the results are presented in Figure 6c. The associated thermodynamic parameters are summarized in Table 4. With an increase in temperature from 298.15 K to 318.15 K, as indicated in Table 2, the *q*_m_ slightly increased from 140.74 mg/g to 143.38 mg/g, suggesting an endothermic adsorption process. Furthermore, the *K*_L_ increased from 0.136 to 0.825, indicating a higher affinity for adsorption at elevated temperatures. It can be observed from Table 4 that the adsorption of RBB KN-R onto CBHBs was a spontaneous process with a negative ΔG. Moreover, the positive ΔH and ΔS values indicated an endothermic adsorption process and increased randomness at the solid–liquid interface.

### 2.7. Effect of Ionic Strength and Coexisting Substances

The impact of ionic strength (NaCl concentration) on the adsorption capacity of RBB KN-R on CBHBs is illustrated in Figure 7a. The results depicted in Figure 7a demonstrate a decrease in adsorption capacity with increasing ionic strength, which is in line with previous studies on the removal of organic pollutants through adsorption [24,41,42]. The decreased adsorption capacity was mainly due to the electrostatic screening effects, that is, the presence of Cl^-^ competed with Dye-SO_3_^−^ for the available adsorption sites on CBHBs, resulting in a decline in adsorption capacity as the ionic strength increased.

Figure 7b exhibited the influence of coexisting substances on the adsorption capacity of RBB KN-R on CBHBs. It can be observed that, compared to Cl^−^, the coexistence of PO_4_^3−^ slightly decreased the adsorption capacity, whereas the coexistence of SO_4_^2−^ significantly reduced the adsorption capacity. Generally, coexisting anions competed with Dye-SO_3_^−^ for adsorption sites, and the competitive effect became more pronounced with higher-valence anions. At pH 2, PO_4_^3−^ predominantly existed as H_3_PO_4_ and HPO_4_^2−^, carrying a lower charge density than SO_4_^2−^. Consequently, the adsorption capacity in the presence of coexisting SO_4_^2−^ was lower than that in the presence of coexisting PO_4_^3−^ [21].

### 2.8. Adsorption Mechanism

The adsorption mechanism of RBB KN-R was multifaceted due to its complex nature. Both intraparticle and film diffusion processes were observed during the adsorption, as indicated by the results of the adsorption kinetic and isotherm analyses. Additionally, the FTIR spectrum before and after adsorption (Figure 2) revealed changes in peak density around 3430 cm^−1^, 1642 cm^−1^, 1575 cm^−1^, and 1078 cm^−1^, suggesting the involvement of oxygen-containing functional groups (–OH, C=O, C–O–C), nitrogen-containing functional groups (–NH_2_), and benzene rings in the adsorption process of RBB KN-R.

To further investigate the adsorption mechanism, XPS analysis was conducted on CBHBs before and after adsorption. The C1s spectra (Figure 8a) and N1s spectra (Figure 8b) of CBHBs exhibited distinct peaks at different binding energies. Specifically, in the C1s spectra of CBHBs before adsorption, the peaks at 284.8, 287.1, 285.9, and 288.6 eV corresponded to C–H, C–O, C–N, and C=O, respectively [16,43]. In N1s spectra of CBHBs before adsorption, the peaks located at 399.6 and 402.5 eV were corresponding to –NH_2_ and N^+^ [44]. Following adsorption, an increase in the C=C peak at 283.9 eV was observed, which could be attributed to the adsorbed RBB KN-R or the π–π interaction between CBHBs and RBB KN-R. Furthermore, there was a decrease in the molar ratio of C–O from 23.39% to 15.35%, C–N from 19.29% to 12.84%, –NH_2_ from 48.95% to 45.23%, and an increase in the molar ratio of N^+^ from 51.05% to 54.77%, indicating that C–O, C–N, –NH_2_, and N^+^ may participate in the RBB KN-R removal. Hence, it can be postulated that hydrogen bonding and electrostatic attractions between CBHBs and RBB KN-R may contribute to the adsorption process.

Overall, based on the experimental and characterization results, there were numerous functional groups on RBB KN-R and CBHBs, such as –SO_3_^−^, –NH_2_ and benzene ring on RBB KN-R, as well as –OH, –NH_2_, –C=C, and benzene ring on CBHBs. Similar to the RBB KN-R adsorption onto polyaniline/bacterial extracellular polysaccharides composite [45], at pH 2, the anions Dye-SO_3_^−^ on RBB KN-R would combine with the cation –NH_3_^+^ on CBHBs through electrostatic attraction. In addition, the –NH_2_ on RBB KN-R would form hydrogen bonds with nitrogen-containing and oxygen-containing functional groups on the surface of CBHBs, acting as H-donor. Moreover, the benzene ring on RBB KN-R could engage in π–π interactions and hydrophobic interaction with the benzene ring and hydrophobic surface of CBHBs, which is consistent with other studies on the removal of organic pollutants with aromatic rings [46,47,48]. In summary, the proposed mechanisms are illustrated in Figure 9.

### 2.9. Desorption and Reusability

From an economic perspective, the regeneration of CBHBs holds great significance. In this study, a desorption solution of 1 M NaOH was employed, and the results are presented in Figure 10. Upon repeated use for 3 cycles, the adsorption capacity of RBB KN-R on CBHBs remained consistently high, exceeding 95% of the initial adsorption capacity. These findings indicated that CBHBs were an effective adsorbent for the removal of RBB KN-R from wastewater, thereby establishing their viability as a reliable and sustainable option.

## 3. Materials and Methods

### 3.1. Chemicals

Rice husk BC (fired for 2 h at 500 °C without oxygen with a heating rate of 10 °C/min) was purchased from Henan Lize Environmental Protection Technology Co., Ltd., (Zhengzhou, China). Chitosan (deacetylation degree 80–95%, viscosity 50–800 mPa·s), glutaraldehyde, and RBB KN-R (pKa < 1) [49,50] were procured from Sinopharm Chemical Reagent Co., Ltd., (Shanghai, China). Other chemicals were of analytical grade and were acquired from Tianjin Damao Chemical Co., Ltd., (Tianjin, China).

### 3.2. Preparation of CBHBs

The CBHBs were synthesized using the following procedure: Initially, 0.4 g of BC was uniformly dispersed in 10 mL of chitosan solution (0.2 g chitosan dissolved in 2% acetic acid). Simultaneously, 1.2 g of span-80 was added to 80 mL of liquid paraffin and thoroughly mixed at a temperature of 333.15 K. Subsequently, the BC suspension was added into the liquid paraffin solution and then mixed, followed by pH adjustment to a range of 9–10; then, 3 mL of glutaraldehyde was added. The resulting mixture was allowed to react for 24 h at 200 rpm and 333.15 K, after which the CBHBs were separated from the solution. To eliminate excess unreacted reagents, the obtained CBHBs underwent sequential washing with hot ethanol and deionized water. The CBHBs were then collected through centrifugation and further utilized after drying in an oven at 423.15 K. Figure 11 illustrates the schematic diagram depicting the synthesis process of CBHBs.

### 3.3. Characterization

The scanning electron microscope (SEM) Supra 55 (Zeiss, Oberkochen, Germany) was employed to analyze the morphology and microstructure of BC and CBHBs before and after adsorption. The surface area and pore volume of CBHBs were determined using the Brunauer–Emmett–Teller (BET) method (TriStar II 3020, Mike, Norcross, GA, USA). Fourier transform infrared spectroscopy (FTIR) analysis (ALPHA, Brooke, Karlsruhe, Germany) was conducted to characterize the functional groups present on the surfaces of BC and CBHBs before and after adsorption. The X-ray photoelectron spectroscopy (XPS) analysis (Nexsa, Thermo Fisher Scientific, Waltham, MA, USA) was performed to investigate the valence state of elements on the surface of CBHBs before and after adsorption with background vacuum of analysis chamber at 1 × 10^−9^ mbar and Al Kα ray excitation source at an energy of 1486.68 eV. The pH-dependent point zero charges (pH_PZC_) of CBHBs were measured using the pH drift method [51].

### 3.4. Adsorption Experiments

The effects of pH, temperature, coexisting substances, and ionic strength were assessed at different pH levels (ranging from 2 to 9), temperatures (298.15 K, 308.15 K, and 318.15 K), coexisting substances at a concentration of 0.04 M (NaCl, Na_3_PO_4_, and Na_2_SO_4_), and NaCl concentrations (from 0 to 0.14 M). The experimental setup consisted of a conical flask with 500 mL of RBB KN-R solution having an initial concentration of 100 mg/L, along with the addition of 0.2 g of CBHBs.

The differences between BC and CBHBs, as well as the impact of CBHBs dosage, were investigated at different BC and CBHBs dosage. The experimental setup involved a 500 mL solution of RBB KN-R with an initial concentration of 300 mg/L; in addition, either 0.2 g of CBHBs or 0.2 g of BC was added to the solution at pH 2 and 298.15 K.

Isotherm studies were performed by conducting adsorption experiments using varying concentrations of RBB KN-R (ranging from 0 to 300 mg/L) at pH 2 and temperatures of 298.15 K, 308.15 K, and 318.15 K. Kinetic studies were carried out by conducting adsorption experiments with different initial concentrations of RBB KN-R (50 and 100 mg/L) over various time intervals at pH 2 and 298.15 K.

Following adsorption, the solution was centrifuged, and the supernatant was collected for RBB KN-R concentration analysis using UV–Vis spectrophotometry at 591 nm [26]. The adsorption capacity (*q_e_*, mg/g) was determined using the following equations:(1)$$q_{e}=\frac{(c_{o}-c_{e})V}{m}$$
where *c_o_* (mg/L) and *c_e_* (mg/L) represent the initial and equilibrium concentrations of RBB KN-R, respectively; *V* (L) denotes the volume of the initial solution; *m* (g) refers to the mass of the adsorbent.

### 3.5. Desorption and Regeneration Studies

After the completion of the adsorption process, the adsorbents were separated from the solution and subsequently washed with deionized water. To desorb the adsorbed RBB KN-R from the CBHBs, a 1 M NaOH solution was employed as the eluent to regenerate the CBHBs. The regenerated CBHBs were utilized in subsequent adsorption experiments for the next cycle.

### 3.6. Statistical Analysis

The Langmuir model and Freundlich model were used to fit the isotherm dates [52]. The pseudo-first-order kinetic model, pseudo-second-order kinetic model, and intraparticle diffusion model were chosen to fit the kinetic dates [21], and the Van’t Hoff equation was used to calculate the thermodynamic parameters [53].

The Langmuir model and separation factor constant (*R_L_*) is expressed as follows:(2)$$q_{e}=\frac{q_{m}K_{L}c_{e}}{1+K_{L}c_{e}}$$
(3)$$R_{L}=\frac{1}{1+K_{L}c_{o}}$$

The Freundlich model is expressed as follows:(4)$$q_{e}=K_{F}c_{e}^{1/n}$$
where *c_o_* (mg/L) and *c*_e_ (mg/L) represent the initial and equilibrium concentration of RBB KN-R in solution, *q_e_* (mg/g) denotes the amounts of RBB KN-R adsorbed on CBHBs at equilibrium, *q_m_* (mg/g) refers to the maximum adsorption capacity, *K_L_* (L/mg) is the equilibrium constant of the Langmuir model, and *K_F_* and 1/n are Freundlich isotherm constants related to adsorption capacity (mg/g) and sorption intensity, respectively. 

The model for the pseudo-first-order kinetic is expressed as follows:(5)$$q_{t}=q_{e}(1-e^{-k_{1}t})$$

The model for the pseudo-second-order kinetic is expressed as follows:(6)$$q_{t}=\frac{k_{2}q_{e}^{2}t}{1+k_{2}q_{e}t}$$

The intraparticle diffusion model is expressed as follows:(7)$$q_{t}=k_{dif}t^{1/2}+\theta$$
where *q*_t_ (mg/g) is the adsorption amount of the RBB KN-R adsorbed on CBHBs when the reaction time is *t*, *θ* (mg/g) is the intercept of the intraparticle diffusion model, and *k*_1_ (min^−1^), *k*_2_ (g/(mg·min)), and *k*_dif_ ((mg/(g·min^−1/2^))) represent the rate constants of the pseudo-first order, pseudo-second order kinetic, and intraparticle diffusion model, respectively.

The Van’t Hoff equation for the thermodynamic parameters calculated is as follows:(8)ΔG=ΔH−TΔS
(9)$$\ln k_{d}=-\frac{\Delta H}{RT}+\frac{\Delta S}{R}$$
(10)$$k_{d}=\frac{q_{e}}{c_{e}}\times 1000$$
where Δ*G*, Δ*H*, and Δ*S* represent the Gibbs free energy, enthalpy, and entropy, respectively, *k_d_* is the distribution coefficient of adsorption, and *T* is the operating temperature in Kelvin.

## 4. Conclusions

In this study, chitosan–biochar hydrogel beads (CBHBs) were successfully synthesized to facilitate the removal of RBB KN-R from wastewater. A pH of 2 was identified as the optimum condition for the efficient elimination of RBB KN-R. The adsorption process was accurately characterized by the pseudo-second-order kinetic model and the Freundlich isotherm model. Furthermore, a combined mechanism involving π–π interactions, electrostatic attraction, hydrophobic interaction, and hydrogen bonding was observed during the removal of RBB KN-R. The presence of coexisting substances and increased ionic strength led to a decrease in the adsorption capacity. Remarkably, CBHBs exhibited sustained high adsorption capacity even after undergoing three cycles of regeneration using a 1 M NaOH desorption solution. These findings provide evidence that CBHBs could serve as an effective and cost-efficient adsorbent for the removal of RBB KN-R.

## Figures and Tables

**Figure 1 molecules-28-06137-f001:**
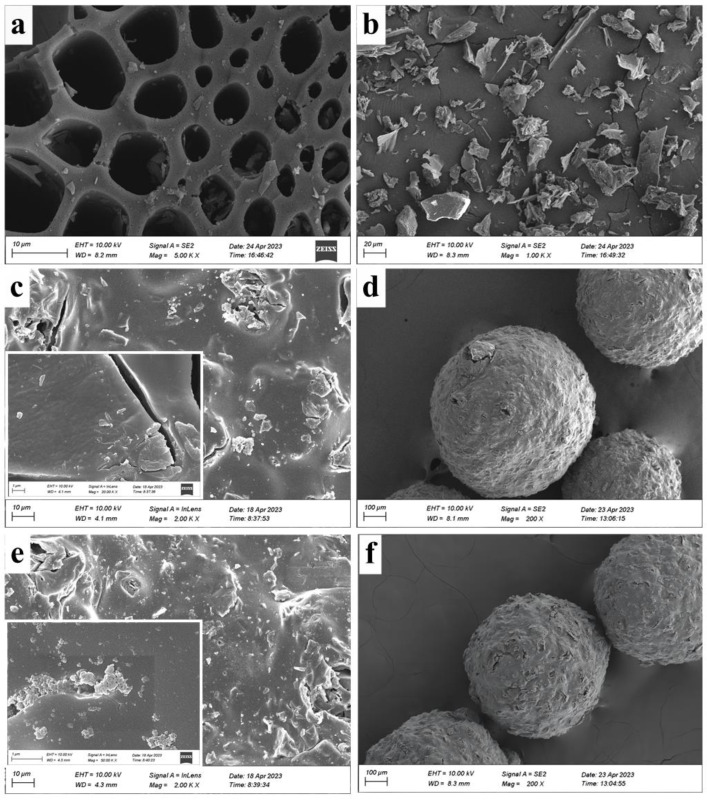
SEM images of BC (**a**,**b**), CBHBs before adsorption (**c**,**d**), and CBHBs after adsorption (**e**,**f**).

**Figure 2 molecules-28-06137-f002:**
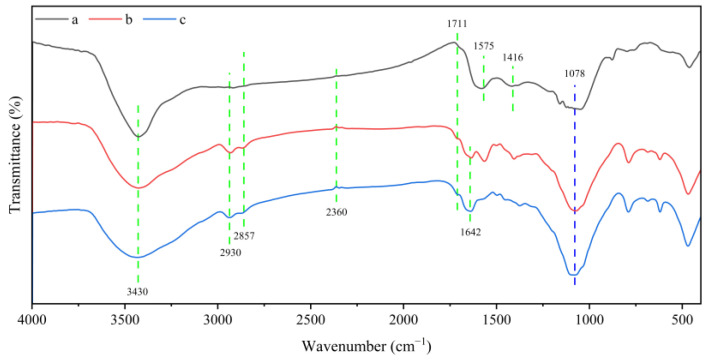
FTIR spectra of (a) biochar; (b) CBHBs before RBB KN-R adsorption; (c) CBHBs after RBB KN-R adsorption.

**Figure 3 molecules-28-06137-f003:**
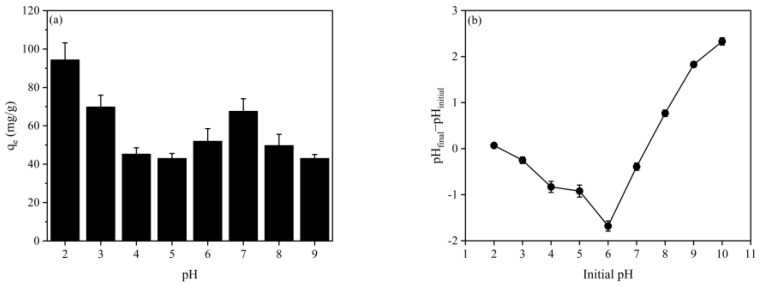
(**a**) Adsorption capacity of RBB KN-R on CBHBs at different pH conditions; (**b**) the point of zero charge of CBHBs.

**Figure 4 molecules-28-06137-f004:**
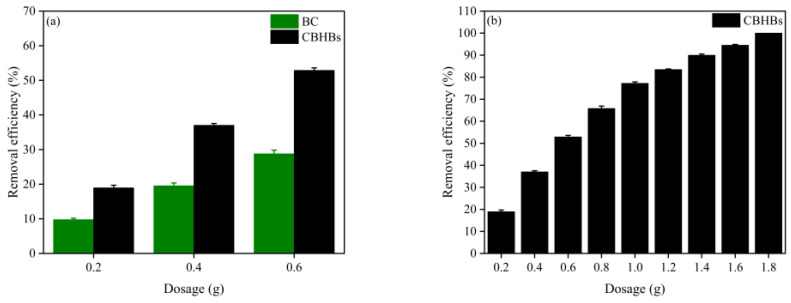
(**a**) RBB KN-R removal efficiency of BC and CBHBs at different adsorbent dosages; (**b**) RBB KN-R removal efficiency of CBHBs at different adsorbent dosages.

**Figure 5 molecules-28-06137-f005:**
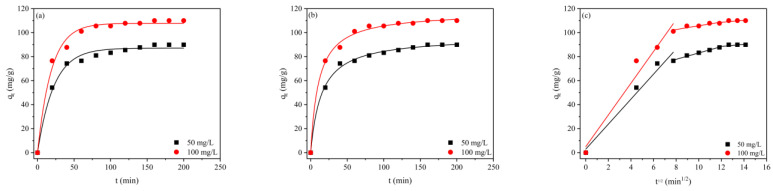
(**a**) Pseudo-first-order kinetic model fitting plots for RBB KN-R adsorption onto CBHBs; (**b**) pseudo-second-order kinetic model fitting plots for RBB KN-R adsorption onto CBHBs; (**c**) intraparticle diffusion model fitting plots for RBB KN-R adsorption onto CBHBs.

**Figure 6 molecules-28-06137-f006:**
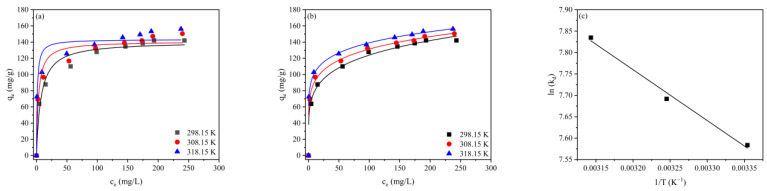
(**a**) Langmuir model adsorption isotherms for RBB KN-R adsorption onto CBHBs; (**b**) Freundlich model adsorption isotherms for RBB KN-R adsorption onto CBHBs; (**c**) the Van’t Hoff plots for RBB KN-R adsorption onto CBHBs at 298.15 K, 308.15 K, and 318.15 K.

**Figure 7 molecules-28-06137-f007:**
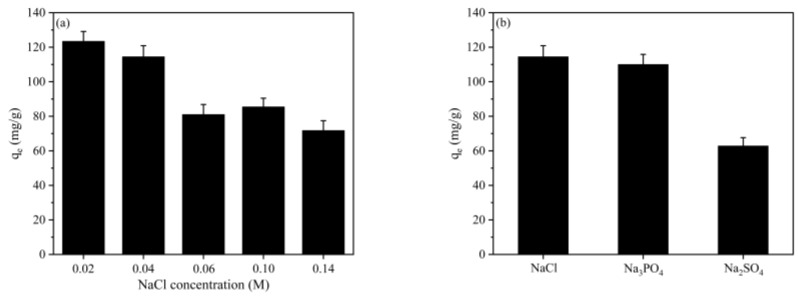
Effect of ionic strength (**a**) and coexisting substances (**b**) on the adsorption capacity of RBB KN-R.

**Figure 8 molecules-28-06137-f008:**
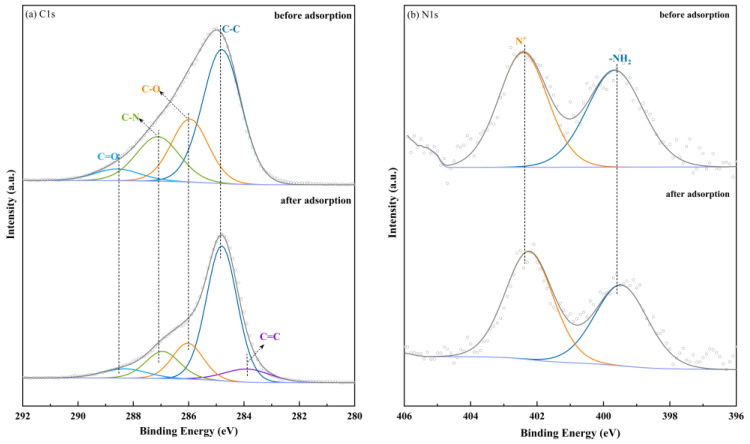
(**a**) XPS C1s core spectra of CBHBs before and after adsorption; (**b**) XPS N1s core spectra of CBHBs before and after adsorption.

**Figure 9 molecules-28-06137-f009:**
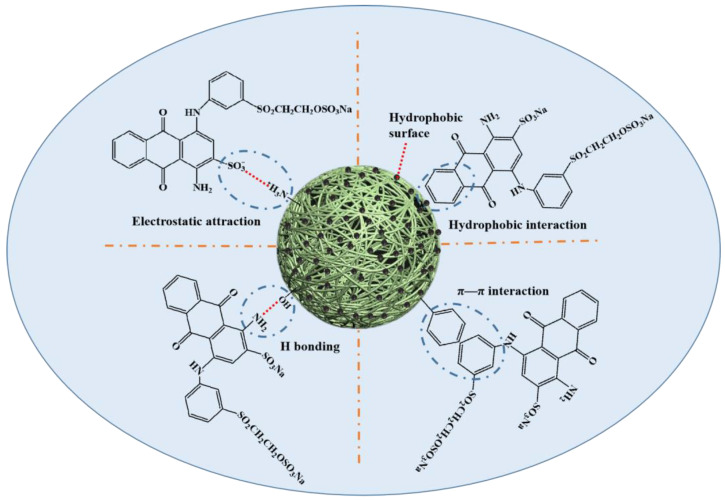
Proposed adsorption mechanism of RBB KN-R on CBHBs.

**Figure 10 molecules-28-06137-f010:**
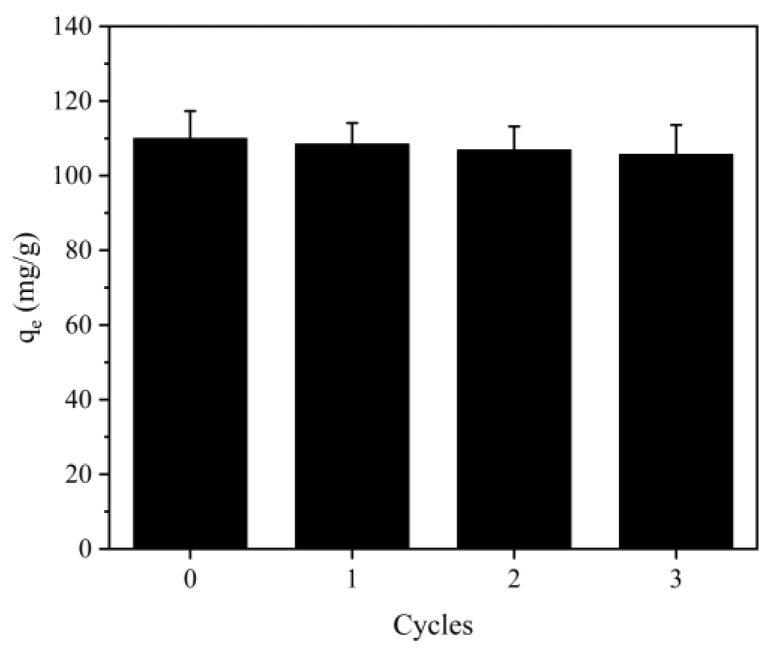
The adsorption capacity of RBB KN-R on CBHBs in different regeneration cycles.

**Figure 11 molecules-28-06137-f011:**
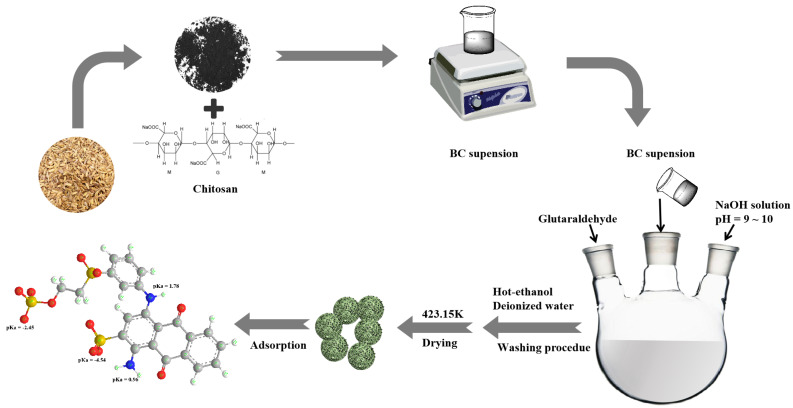
Schematic diagram for preparation of CBHBs.

**Table 1 molecules-28-06137-t001:** Applied adsorption kinetic equations and kinetic parameters for the adsorption of RBB KN-R onto CBHBs.

*c*_0_ (mg/L)	*q*_e,exp_ (mg/g)	Pseudo-First-Order Kinetic Model			Pseudo-Second-Order Kinetic			Intraparticle Diffusion Model		
		*q*_e,cal_ (mg/g)	*k*_1_ (min^−1^)	R^2^	*q*_e,cal_ (mg/g)	*k*_2_ (g/(mg·min))	R^2^	*k*_f_ (mg/(g·min^−1/2^))	*θ* (mg/g)	R^2^
50	89.95	87.11	0.0451	0.9875	96.90	6.970 × 10^−4^	0.9968	2.645	56.64	0.9830
100	110.04	107.82	0.0531	0.9883	117.23	7.731 × 10^−4^	0.9967	1.564	90.10	0.8028

**Table 2 molecules-28-06137-t002:** Isotherm parameters of RBB KN-R adsorption onto CBHBs at various temperature.

T (K)	Langmuir				Freundlich		
	*K*_L_ (L/g)	*q*_m_ (mg/g)	*R* _L_	R^2^	*K*_F_ (mg^(1−n)^ L^n^/g)	1/n	R^2^
298.15	0.136	140.74	0.8803	0.9714	50.57	0.1947	0.9864
308.15	0.307	140.98	0.7651	0.9532	63.18	0.1590	0.9932
318.15	0.825	143.38	0.5479	0.9318	74.65	0.1352	0.9975

**Table 3 molecules-28-06137-t003:** Comparative analysis of the maximum adsorption capacity of RBB KN-R by different adsorbents as reported in the literature.

Adsorbent	*q*_m_ (mg/g)	Reference
Mesoporous activated carbon from industrial laundry sewage sludge	33.47	[26]
*T. conoides*-derived biochar	92.5	[35]
Active *S. quadricauda*/alginate (IASq)	68	[36]
Heat inactivated *S. quadricauda*/alginate (IHISq)	95.2	[36]
Mg(OH)2-bentonite	66.90	[37]
CHT-GLA/ZnO/Fe_3_O_4_	176.6	[38]
Chitosan–alkali lignin composite	111.11	[39]
Chitosan grafted-benzaldehyde/montmorillonite/algae	213.6	[40]

**Table 4 molecules-28-06137-t004:** The thermodynamic parameters calculated for the adsorption of RBB KN-R, from an initial RBB KN-R concentration of 100 mg/L, 0.2 g CBHBs, and pH 2.

T (k)	Δ*G* (KJ/mol)	Δ*H* (KJ/mol)	Δ*S* (KJ/mol)
298.15	−18.77	9.883	0.0961
308.15	−19.73		
318.15	−20.69		

## Data Availability

The data are contained within the article.

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
