# Peer review of "Enhanced Reactive Brilliant Blue Removal Using Chitosan–Biochar Hydrogel Beads"

_molecules, 2023, doi:10.3390/molecules28166137_

Round 1

Reviewer 1 Report

This paper focuses on preparation of the chitosan-biochar hydrogel beads, characterization of their physicochemical parameters as well as their potential application for organic dyes removal from wastewater. Generally speaking the article is interesting, especially from practical and ecological points of view. However, the quality of the work in its present form is not suitable for publication in Molecules. There is a large number of flaws that should be reviewed, and redone before the manuscript has been considered for publication.

The detailed comments are specified below:

1.     Please change the “RBB KN-R” keyword.

2.     The description of biochar synthesis methods in the Introduction should be slightly extended. Please provide information on methods and thermal conditions for the synthesis of such materials.

3.     The layout of the manuscript should be changed according to the instructions for authors. The experimental part should be after the discussion of the results.

4.     Please provide information on rice husks pyrolysis conditions.

5.     Lines 81-82 and 84: Please explain this sentence: “Initially, 0.4 g of BC was uniformly dissolved in 10 ml of chitosan solution…” Was the biochar dissolved or rather dispersed in the chitosan solution? The same applies to the statement “Subsequently, the BC solution was mixed with…”.

6.     Does the procedure for washing excess unreacted reagents have to be so extensive? Washing five times with acetone, ethanol, and deionized water is not very beneficial, especially from an ecological point of view.

7.     What was the grain size of the prepared composite chitosan-BC hydrogel beads?

8.     Please prepare graphic illustration of the proposed synthesis procedure for composite chitosan-BC hydrogel beads.

9.     Please specify the conditions for degassing the samples before textural measurements.

10.  The impact of the adsorbent dose on the efficiency of dye adsorption should also be investigated.

11.  Lines 154-155: Please clarify the statement: “…whereas the surface of CBHB exhibited roughness and the presence of numerous pore channels and internal mesopores”. How can the presence of mesopores with sizes of 2-50 nm be determined based on the SEM (scale in micrometers). It's a completely different order of magnitude. Moreover, in order to accurately compare the materials (before and after adsorption), SEM images of the same scale should be included.

12.  Lines 156: The size of the specific surface area should be rounded to a whole number or to 1 decimal place. The error of this measurement method is usually 2-5%. Total pore volume should be also provided.

13.  Discussion of FT-IR results may be omitted or included in supplementary files. The surface chemistry (type and amount of functional groups) can be discussed on the basis of the XPS studies conducted by the authors, which are more authoritative.

14.  Please add error bars in the Figures.

15.  Analogous adsorption tests should be carried out for the initial biochar in order to show the effectiveness of the obtained composite.

16.  The prepared composite chitosan-BC hydrogel beads should be also compared with other adsorbents described in the literature (in a form of Table).

Taking into account all the above objections, I do not recommend publication of this paper, at its present form. A major revision is needed.

Reviewer 2 Report

The manuscript by Zhao et al. devoted to research of dye adsorption on biochar-chitosan composite hydrogel. A lot of techniques was employed in this research, and authors can show that proposed adsorbent is effective for brilliant blue removing. However, manuscript contains some flows and I have some questions and recommendations listed below:

line 53: typo “acetamidos-D-glucose”, also chitosan is deacetylated

line 65-67: “In this study, composite chitosan-BC hydrogel beads (CBHB) were fabricated by encapsulating BC within chitosan to enhance mechanical strength, stability, adsorption performance, and recycling ability for the removal of RBB KN-R. Rice husk was subjected to pyrolysis to produce BC.” the authors didn’t research mechanical strength of hydrogel obtained, so this phrase should be modified.

2.1 Chemicals: what are chitosan characteristics? Molecular weight and degree of deacetylation should be provided. Also, the structure and pKa value(s) of RBB KN-R should be provided in the manuscript, not only in the its end in Fig. 8.

line 81: “0.4 g of BC was uniformly dissolved in 10 ml of chitosan solution…” Is BC soluble in acidic medium? May be BC was dispersed in chitosan solution?

Fig. 1: all images have different magnification, so they can’t be compared. The authors should provide the SEM images with the same magnification. Also, Fig. 1B clearly demonstrates porous structure of BC.

Figs. 3-9: error bars should be demonstrated.

line 200: why pi-pi interactions only? H-bonds and hydrophobic interactions may also take part in adsorption of dye on bead surface. The RBB KN-R parameters should be provided to confirm author’s suggestions.

line 305: “… indicating hydrogen bonding and electrostatic attractions between CBHB and RBB KN-R” the data presented shows only interactions between components. The type of interactions is not clear, so, I recommend rephrase this sentence.

 Minor editing of English language required

Round 2

Reviewer 2 Report

The authors made required corrections, so manuscript can be accepted in the present form.

English needs minor corrections